# Beyond Fine-Tuning: The Present and Future of Parameter-Efficient Fine-Tuning in Vision Transformers

Edwin Kwadwo Tenagyei
Griffith University
Brisbane, Queensland, Australia
edwinkwadwo.tenagyei@griffithuni.edu.au

Lei Wang*
Griffith University
Brisbane, Queensland, Australia
Data61/CSIRO
Canberra, Australian Capital
Territory, Australia
l.wang4@griffith.edu.au

Yongsheng Gao*
Griffith University
Brisbane, Queensland, Australia
yongsheng.gao@griffith.edu.au

## Abstract

The advent of Vision Transformers (ViTs) has significantly advanced computer vision, particularly image classification, through large-scale pretraining on massive datasets. However, the high computational and memory costs of full fine-tuning remain a major bottleneck for practical deployment across diverse downstream tasks. Parameter-Efficient Fine-Tuning (PEFT) has emerged as a promising paradigm to address this challenge by adapting models through updates to only a small subset of parameters while preserving the benefits of pretrained representations. In this survey, we present a focused review of PEFT techniques for ViTs in image classification. We introduce a structured taxonomy that categorizes existing approaches into additive-based, reparameterization-based, selective, hybrid, and inference-efficient tuning methods, highlighting their core design principles, strengths, and limitations. We further analyze evaluation protocols, benchmark results, and trade-offs between accuracy, efficiency, and scalability. Finally, we identify open challenges and outline promising directions for future research toward more robust, efficient, and deployable PEFT frameworks for vision. Our appendix is available here.

## CCS Concepts

• **General and reference** → **Surveys and overviews**; • **Computing methodologies** → *Machine learning*; *Computer vision*.

## Keywords

parameter-efficient, fine-tuning, vision transformers, pre-training

**ACM Reference Format:**
Edwin Kwadwo Tenagyei, Lei Wang, and Yongsheng Gao. 2026. Beyond Fine-Tuning: The Present and Future of Parameter-Efficient Fine-Tuning in Vision Transformers. In *Companion Proceedings of the ACM Web Conference 2026 (WWW Companion '26), April 13–17, 2026, Dubai, United Arab Emirates.* ACM, New York, NY, USA, 14 pages. https://doi.org/10.1145/3774905.3794673

*Corresponding authors.

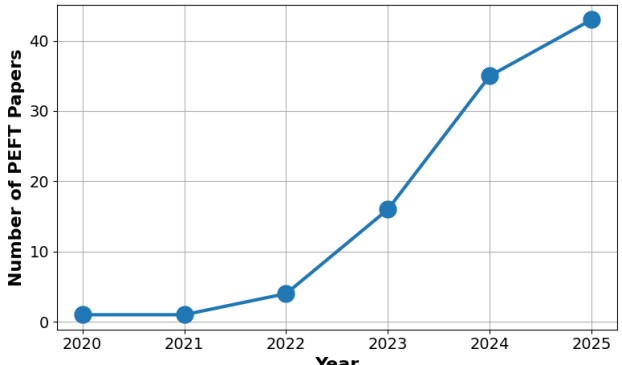

**Figure 1: Illustrative growth trend of publications on parameter-efficient fine-tuning for Vision Transformers in image classification (2020-2025), based on a curated set of representative works.**

## 1 Introduction

The remarkable success of deep learning in computer vision has been driven by the development of large-scale pre-trained models. Among these, Vision Transformer (ViT) [17] have emerged as a dominant architecture, demonstrating strong performance on image classification datasets[14, 79] and a wide range of downstream tasks[48, 74, 105, 107]. By modeling long-range dependencies through self-attention mechanisms, ViTs exhibit competitive and often generalization compared to CNNs, particularly when trained on massive datasets such as ImageNet[14, 79]. As a result, ViTs have become the backbone of many state-of-the-art vision systems leading to a large number of transformer-based pre-trained vision models (PVMs)[7, 17, 28, 58, 90, 100].

PVMs have demonstrated impressive representational capabilities and have emerged as the de facto approach for fine-tuning models on a wide range of downstream tasks. However, full fine-tuning of PVMs across several downstream tasks remains computationally expensive[4, 73]. The process involves updating hundreds of millions of parameters, which demands significant GPU memory, training time, and energy consumption. These costs pose significant challenges in resource-constrained settings, such as on-device deployment or scenarios requiring adaptation to multiple downstream tasks with limited labeled data.

 Edwin Kwadwo Tenagyei, Lei Wang, and Yongsheng Gao

To address these limitations, Parameter-Efficient Fine-Tuning (PEFT), originally from from the field of natural language processing (NLP) [34], has emerged as a compelling alternative. Rather than updating the entire model, PEFT methods adapt only a small subset of parameters while keeping the pretrained backbone largely frozen. This strategy significantly reduces training cost, improves scalability across tasks, and frequently achieves performance comparable to full fine-tuning. Recent works[34, 35, 38, 106] has introduced a diverse range of PEFT techniques for ViTs, inserting additional learnable parameters [38], lightweight additional modules [9, 10, 34], and attention weight decomposition [35]. Beyond efficiency gains, these approaches also offer insights into the representational capacity and adaptability of ViTs.

The growing interest in PEFT for vision is also reflected in the rapid increase in related research activity. As illustrated in Figure 1, the number of publications focusing on parameter-efficient fine-tuning for ViTs in image classification has risen sharply in recent years. This trend, based on a curated set of representative works, highlights both the maturation of PEFT techniques in vision, and the increasing demand for efficient adaptation strategies as model scales continue to grow. At the same time, the accelerating pace and diversity of research underscore the need for a structured synthesis of existing methods and findings. Although several studies investigate PEFT in diverse vision and multimodal settings [6, 86, 108], a focused synthesis dedicated to PEFT for ViTs in image recognition remains limited. Image recognition is a foundational task in computer vision and serves as a primary evaluation benchmark for assessing the effectiveness of PEFT methods in vision. Given the rapid growth and diversification of this research area, there is a clear need to consolidate existing work, analyze emerging trends, and identify open challenges.

In this work, we present a focused survey on PEFT for ViTs in image recognition tasks. Our main **contributions** are:

i. We propose a structured taxonomy of PEFTs for ViTs, covering additive-based tuning, reparameterization-based tuning, selective tuning, hybrid-based tuning and inference-efficient tuning.
ii. We review benchmark results, evaluation protocols, and trade-offs, providing insights into the efficiency, scalability, and performance characteristics of different PEFT paradigms.
iii. We discuss open research challenges and outline promising future directions for PEFT in image classification, with particular emphasis on robustness, inference efficiency, and scalable deployment.

## 2 Preliminaries

### 2.1 Vision Transformer

The Vision Transformer (ViT) is composed of a patch embedding module followed by $L$ transformer encoder layers. Given an input image $x \in \mathbb{R}^{H \times W \times C}$, the patch embedding splits $x$ into a sequence of patches and flattens them into $x_p \in \mathbb{R}^{N \times (P^2 C)}$, where $(H, W)$ denote the image height and width, $(P, P)$ is the patch resolution, $C$ is the number of channels, and $N = HW/P^2$ is the total number of tokens. These patch vectors are then projected through a learnable linear layer to obtain $x_0 \in \mathbb{R}^{N \times d}$. A special classification token [cls] is prepended to $x_0$, and the resulting sequence forms the input to the transformer encoder.

Each transformer layer consists of a Multi-Head Self-Attention (MHSA) module and a Multilayer Perceptron (MLP). In MHSA, attention weights are computed using query ($Q$), key ($K$), and value ($V$) matrices derived from the input $x_{\ell-1}$ at layer $\ell$ with projection parameters $W_q, W_k, W_v \in \mathbb{R}^{d \times d}$:

$$Q = x_{\ell-1} W_q, \quad K = x_{\ell-1} W_k, \quad V = x_{\ell-1} W_v, \tag{1}$$

$$x'_\ell = \text{Attention}(Q, K, V) = \text{softmax}\left(\frac{QK^\top}{\sqrt{d}}\right) V. \tag{2}$$

The output $x'_\ell$ is then normalized and passed through an MLP block with a residual connection:

$$x_\ell = \text{MLP}(\text{LN}(x'_\ell)) + x'_\ell, \tag{3}$$

where $x_\ell$ is the output of the $\ell$-th layer and LN is Layer-Norm.

### 2.2 Transfer Learning

Transfer learning plays a major role in modern machine learning. It is designed to harness the knowledge embedded within a large pretrained model to enhance the performance of downstream tasks [89, 99, 122, 124]. At the core, the parameters $\theta$ of a pretrained large vision models encode rich linguistic and contextual knowledge. These parameters can be efficiently adapted to downstream tasks either by fine-tuning a small subset of them or by introducing lightweight task-specific components[34, 35]. This paradigm highlights the effectiveness of leveraging broad pretraining to achieve strong task-specific performance.

Let us denote a pretrained model $f_{\theta_0}$ with parameters $\theta_0$, obtained through training on a large-scale image dataset. Transfer learning allows the adaptation of this model to a downstream dataset $\mathcal{D}_{\text{task}}$. The adaptation process seeks an optimized model $f_{\theta^*}$, where the optimization objective is given by:

$$\theta^* = \arg\min_\theta \mathcal{L}(f_\theta, \mathcal{D}_{\text{task}}), \tag{4}$$

with $\mathcal{L}$ representing the task-specific loss function. The pretrained parameters $\theta_0$ serve as a strong initialization, improving generalization and acting as a form of regularization. This significantly reduces the amount of task-specific training required, making adaptation both computationally efficient and effective. The principles of transfer learning form the foundation of PEFT methods.

Inspite of its major role, training and fine-tuning of large vision models impose huge computational and memory costs due to the self-attention mechanism. The time complexity of self-attention is $O(n^2)$ with respect to the number of tokens $n$, since every token attends to all other tokens. During pretraining, large vision models process large number of tokens, and this quadratic cost translates into trillions of floating-point operations (FLOPs)[4, 73]. Fine-tuning exacerbates the challenge, as it typically requires updating all parameters for each downstream task, especially when dealing with complex datasets. Additionally, the memory requirements of large vision models are similarly demanding. Storage scales with both the number of model parameters and the size of intermediate activations, which grows on the order of $O(n \cdot d_{\text{model}})$, where $d_{\text{model}}$ is the hidden dimension of the model. During training, memory consumption includes storing parameters as well as their gradients, leading to significant memory overhead, particularly when all parameters are updated during full fine-tuning.

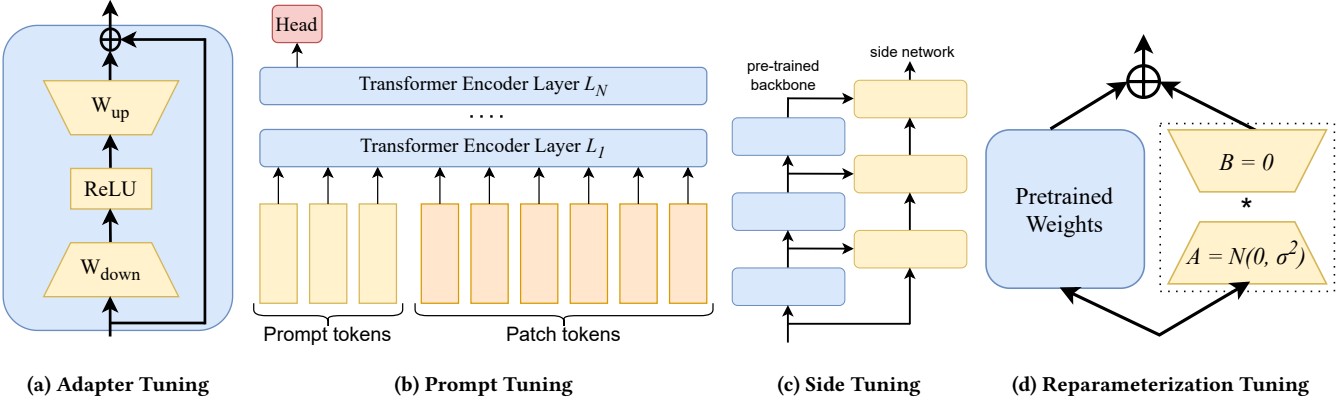

**(a) Adapter Tuning**  **(b) Prompt Tuning**  **(c) Side Tuning**  **(d) Reparameterization Tuning**

**Figure 2: Detailed architectures of representative PEFT methods for Vision Transformers. Adapter tuning modifies internal representations via bottleneck layers; prompt tuning adapts the model through learnable input tokens; reparameterization tuning applies constrained parameter transformations to pretrained weights; and side tuning learns task-specific representations through an auxiliary network while keeping the backbone frozen.**

PEFT techniques offer a practical solution to these challenges by modifying only a small fraction of the parameters. As a result, PEFT aims for a faster training and deployment on resource-limited hardware while maintaining strong task performance.

## 2.3 Parameter Efficient Fine-Tuning

PEFT methods aim to adapt pretrained models to new tasks by updating only a small subset of parameters, rather than the entire model. Consider a pretrained model $M$ with parameters $\theta$, and a downstream task $\mathcal{D} = \{(x_i, y_i)\}_{i=1}^{|\mathcal{D}|}$, where $(x_i, y_i)$ denotes an input-output pair from $\mathcal{D}$. PEFT seeks to adapt $\theta$ to the target task by introducing task-specific parameter increments $\Delta\theta$, such that $|\Delta\theta| \ll |\theta|$. The objective is to optimize these additional parameters by minimizing the task-specific loss $\mathcal{L}$:

$$\min_{\Delta\theta} \ \mathbb{E}_{(x_i,y_i)\in\mathcal{D}} \left[ \mathcal{L}(M_{\theta+\Delta\theta}(\hat{y}_i \mid x_i), y_i) \right]. \tag{5}$$

## 3 Taxonomy

This section presents a comprehensive taxonomy of PEFT methods for image classification tasks. We categorize PEFTs into additive-based tuning, reparameterization-based tuning, selective tuning , hybrid tuning and inference-efficient tuning methods.

### 3.1 Additive-based Tuning

Additive-based tuning strategies introduce supplementary trainable modules or parameters into pre-trained vision models, enabling the model to acquire task-specific knowledge. This class of methods is commonly divided into four main categories: adapter, prompt, prefix, and side tuning.

**Adapter fine-tuning.** The adapter mechanism was first proposed in the NLP domain as an effective approach to PEFT [34]. Owing to its strong performance, it has since been widely adopted in computer vision. The core idea is to insert lightweight neural modules, referred to as *adapters*, into the transformer layers. During adaptation, only these added modules are updated while

the backbone remains frozen. Each adapter consists of a down-projection layer, parameterized by $W_{\text{down}} \in \mathbb{R}^{d\times k}$, followed by an up-projection layer, parameterized by $W_{\text{up}} \in \mathbb{R}^{k\times d}$, as illustrated in Figure 2a. The bottleneck dimension $k$ (with $k \ll d$) compresses the original representation into a lower-rank space. A ReLU activation is applied between the two linear layers to introduce nonlinearity. For an input feature map $x_\ell \in \mathbb{R}^{N\times d}$, the adapter produces an updated representation as:

$$\hat{x}_\ell = \text{ReLU}(x_\ell W_{\text{down}})W_{\text{up}}, \tag{6}$$

where $W = [W_{\text{down}}; W_{\text{up}}^{\top}] \in \mathbb{R}^{d\times 2k}$ denotes all trainable parameters within the adapter. Building on the concept of the original adapter several variants have been developed, e.g., Adaptformer[10] moves away from the conventional way of adapter mechanism and use a parallel insertion of adapters in the pretrained model. Convpass[40] identifies that the current adapters lack strong inductive biases constraining their performance and therefore proposes an integration of trainable convolutional blocks into the adapter architecture. Steitz and Roth [83] conducted a systematic analysis of adapters, examining their internal structure and implementation choices. They identify key limitations that explain why standard adapters often underperforms relative to alternatives such as low-rank adaptation.

Other methods[15, 16, 42] also focused on optimizing the adapter architecture. Through a quantization-aware training strategy, Jie et al. [42] showed an adapter can be made more storage-efficient by exploiting the robustness of adapter parameters to noise and low numerical precision. Adapter Re-Composing (ARC) [16] introduces the idea of sharing the bottleneck operation's up and down-projections across layers and employs low-dimensional re-composing coefficients to create layer-adaptive adapters. For a flexible bottleneck dimensionality in the adaptations, Dong et al. [15] uses Householder matrices to construct Householder transformation-based adaptations to enhance their efficiency and effectiveness. Furthermore, to improve the efficiency and generalization trade-off in the adapter mechanism, Li et al. [52] jointly incorporates parameter sharing, dynamic token-level allocation, and block-specific designs

within a unified framework.To unleash the potential of each parameter in the adapters, Mixture of Sparse Adapters(MoSA) [115] decomposes the standard adapter into multiple non-overlapping modules, stochastically activates these modules for sparse training, and subsequently merges them into a complete adapter posttuning.

**Prompt tuning.** Visual prompt tuning (VPT) [38] offers an alternative to inserting trainable modules directly into transformer architectures. Instead of modifying the model layers, this approach augments the original input, either the image itself or its embedding by attaching learnable visual prompts as depicted in Figure 2b. These prompts take the form of additional trainable parameters that can be flexibly optimized for a given task. The central objective is to use these task-specific prompts to adjust the input distribution so that it better aligns with the distribution seen during pre-training. The original VPT architecture, consists of two variants: VPT-Shallow and VPT-Deep. In VPT-Shallow, a set of $l$ learnable prompts, denoted as $P = [P_1, P_2, \ldots, P_l] \in \mathbb{R}^{l \times d}$, is inserted into the embedding space of the input patch tokens $x_0 \in \mathbb{R}^{N \times d}$. These prompts are concatenated with the original patch embeddings to produce the final input:

$$x_0 = \text{concat}(P, x_0) = [P, x_0] \in \mathbb{R}^{(l+N) \times d}, \quad (7)$$

where $[\cdot, \cdot]$ denotes concatenation along the token dimension.

VPT-Deep on the other hand injects prompts into the input of *every* transformer layer. During fine-tuning, only these prompt parameters are updated while all pre-trained weights remain frozen. The computational overhead of VPT-Deep depends on the prompt length and token embedding dimension, with empirical studies showing that increasing the prompt length typically yields better downstream performance. Building upon the success of VPT, several variants have been explored. Han et al. [27] and Yoo et al. [109] further enhanced VPT by implementing cross-layer prompt connections with dynamic gating mechanisms. Subsequent studies also enhanced VPT's abilities through convolutional prompts [92], lightweight prompt blocks composed of three convolutional layers [67], better prompt positioning [96] and spatial selection mechanisms for coordinating attention between image patches and visual prompts [72]. Recently, Ren et al. [78], and Park and Chung [70] guide prompt tokens to align with semantically informative regions of the ViT embedding space. Different from the original concept of VPT, other approaches also integrates task-specific prompt at the pixel level,thus integrating these prompts with input images. Wu et al. [103] improves the approach by shrinking the input images, applying data augmentations, and padding the surrounding area with prompt information, enriching the input representation. Diversity-Aware Meta Visual Prompt (DAM-VP) [36] takes a divide-and-conquer approach by segmenting high-diversity datasets into smaller subsets and learning separate prompts for each subset. This strategy effectively addresses the challenges posed by the large diversity of data.

**Prefix fine-tuning.** Different from prompt tuning, prefix tuning [53] introduces learnable prefix matrices in the MHSA mechanism of pretrained vision transformers. Prefix-tuning introduces trainable prefix matrices into the attention mechanism. In particular, it uses two learnable matrices $P_k, P_v \in \mathbb{R}^{l \times d}$, which are prepended to the keys and values of the multi-head attention. This modification

updates the attention formulation in Eq. (3) to:

$$\text{Attention}(Q, K, V) = \text{softmax}\left(\frac{Q[P_k, K]^\top}{\sqrt{d}}\right)[P_v, V]. \quad (8)$$

Although effective, initializing these prefix matrices randomly may introduce undesirable noise that hinders stable convergence during fine-tuning. To overcome this limitation, PATT [110] introduces a parallel attention pathway that avoids random initialization. It generates prefix matrices using two linear layers,parameterized by $W_{\text{down}} \in \mathbb{R}^{d \times k}$ and $W_{\text{up}} \in \mathbb{R}^{k \times l}$,together with Tanh activations to transform the input features. For the $\ell$-th transformer layer, given the output of the previous layer $x_{\ell-1}$, the corresponding prefix matrices are computed as:

$$P_k, P_v = \tanh(x_{\ell-1} W_{\text{down}}) W_{\text{up}}. \quad (9)$$

To extend the studies of prefix tuning, VQT [93] appends prefix vectors exclusively to the query Q, rather than increasing both the key K and the value V.

**Side tuning.** Aside parameter efficiency, to improve the memory efficiency of previous PEFT methods, side tuning takes an alternative approach by using a smaller, detached side network running alongside the pretrained vision models. An architecture of side-tuning is shown in Figure 2c. Earlier side tuning approach [114] employed a four-layer convolutional network as an additive side network, whose outputs are combined with representations of the pretrained vision model in the final layer to address various tasks. Recently, several variants [22, 55, 65, 85, 87, 88] of side-tuning have been proposed with the aim of optimizing its architecture.

## 3.2 Reparameterization-based Tuning

Reparameterization-based tuning transforms the model parameters into a lower-dimensional representation during training to facilitate efficient optimization. The reparameterized weights are later mapped back to the original parameter space during inference as shown in Figure 2d, to ensure the model's full capacity and also preserve expressiveness. A popular technique is LoRA [35] which factorizes weight updates into low-rank matrices, significantly reducing the number of trainable parameters. For a pre-trained weight matrix $W_\ell$, LoRA models its adaptation through a low-rank decomposition. Specifically, the updated weight is expressed as

$$W_\ell' = W_\ell + \Delta W = W_\ell + BA, \quad (10)$$

where $B$ and $A$ are the trainable low-rank matrices introduced during fine-tuning. Typically, LoRA modifies the query and value projection matrices in multihead attention. Furthermore, an expanding line of subsequent research has sought to enhance and the LoRA framework. Recent studies have explored the limitations of low rank constraints and proposed several solutions [1, 29, 37, 106, 123], e.g., through Kronecker decompositions which is structurally related to LoRA as shown below:

$$\Delta W = B \otimes A. \quad (11)$$

The fine-tuning process is able to capture more complex relationships utilizing the Kronecker product to constructs full rank update matrices from smaller trainable factors. KAdaptation [29] broadens this idea by expressing the weights as a sum of $n$ Kronecker products, pairing shared slow weights $A_i$ with task-specific fast weights $B_i$. SinLoRA [37] uses a sine function parameterized by

a frequency parameter on top of a low-rank update, effectively producing a full rank update. For products with high effective rank than the original LoRA framework, [1] uses the Khatri-Rao product to produce weight updates. FacT[41] also proposes a tensorization-decomposition framework in which the weights of PVMs are first tensorized into a unified three-dimensional structure and the corresponding parameter updates are then decomposed into compact factorized components enabling an efficient storage of weight increments. Another line of research also investigates the initialization of projection matrices in the LoRA framework [64, 82, 84, 98]. To further enhance transfer representations, other studies have also explored multiple parallel layers [8, 18–20, 47, 121]. Recently, some works [75, 76] proved that there exists a small discrepancy between the pretrained and the finetuned model, limiting the preservation of pretrained knowledge leading to the proposition of orthogonal finetuning (OFT) and its variants [57, 60, 102]. OFT preserves pretrained semantics by mapping pretrained linear weights with an angle-preserving transformation using a block-diagonal matrix, while adapting to downstream tasks. Beyond modifying pre-trained weight matrices, several studies have investigated alternative parameterization strategies within PVMs [54, 80, 81, 113]. For instance, SSF [54] introduces learnable scale and shift parameters to modulate intermediate feature representations, which are later reparameterized into the MLP layers.

### 3.3 Selective Tuning

Selective Tuning adapts pre-trained models to specific taks by modifying only a carefully chosen subset of parameters while keeping the rest unchanged. Unlike traditional fine-tuning, which updates all parameters in the model, selective fine-tuning focuses on parameters that are most relevant to the task. Linear Probe [49] adds a simple linear classifier on top of a frozen pretrained vision model, keeping all pre-trained parameters fixed. Furthermore, BitFit [112] demonstrates empirically that updating only the bias terms of a model can achieve strong fine-tuning performance, highlighting the surprising effectiveness of minimal parameter updates. Instead of updating bias terms, LN-Tune [2] adapts pre-trained models by fine-tuning only their LayerNorm parameters. Another innovative approach is Salient Channel Tuning [119], which adopts a selective channel tuning strategy. This method prioritizes tuning task-relevant channels, thereby significantly reducing parameter costs while maintaining performance. Different from original concept of selective tuning, Zhang et al. [118], and Chen et al. [11] introduce a novel paradigm in which parameters chosen for fine-tuning are determined by their gradient relevance to the downstream task.

### 3.4 Hybrid-based Tuning

Hybrid-based tuning approaches integrate multiple fine-tuning strategies into a unified framework that exploits the complementary advantages of each technique. By combining these mechanisms, hybrid methods offer greater flexibility, adaptability, and robustness across a wide range of tasks. Such approaches can dynamically select or balance the most effective tuning components to achieve strong performance while preserving parameter efficiency. For example, NOAH [117] combines Adapter, LoRA, and VPT mechanisms within each transformer block, using neural architecture search

(NAS) to automatically determine the most effective configuration for a given downstream task. This framework illustrates a holistic approach to fine-tuning, demonstrating how the integration of multiple PEFT techniques can yield optimized adaptation performance. Similarly, V-PEFT [111] provides a unified analysis of PEFT techniques, focusing on critical fine-tuning positions and providing a cohesive perspective on these approaches. In contrast, U-Tuning [39] uncovers a shared parallel structure among mainstream tuning strategies, including adapter, prefix, and prompt tuning, and leverages this insight to reduce structural coupling across these methods. More recently, PEFT-Vision [61] provides a comprehensive analysis of PEFT techniques in vision, offering practical guidance on selecting and applying these methods effectively for different tasks and deployment scenarios.

### 3.5 Inference-Efficient Tuning

Despite the notable progress achieved by current PEFT methods, emerging research [59] shows that most PEFTs such as Adapters inccurs non-negligible latency during inference, even though they significantly reduce the number of trainable parameters. Luo et al. [59] through structural parameterization embeds adapter modules into pre-trained models, ensuring efficient adaptation without zero inference overhead.

In recent years, there has been a growing class of parameter-efficient adaptation methods that reduce the computational burden of ViTs by compressing, selecting, or merging image tokens during fine-tuning [45, 56, 116, 120], inspired by token reduction and token merging techniques. These approaches [5, 46, 77, 91] not only perform parameter efficiency, but also operate on the token dimensions, aiming to retain essential semantic information, eliminate redundancy among spatial tokens thereby improving inference efficiency. Token reduction [46, 77] removes less informative tokens entirely during inference or training, whiles token merging [5, 91] combines similar tokens into a single representative token, preserving information while reducing token count. For instance, DyT [120] incorporates a token dispatcher within each transformer block that dynamically determines which tokens to activate or deactivate. Activated tokens pass through both the full transformer block and an auxiliary lightweight adapter, whereas deactivated tokens skip the block entirely and are processed solely by the adapter. Recently, Kim et al. [45] addresses the inference latency and computational overhead by introducing a plug-and-play token redundancy reduction module that learns token similarity via adapters and performs fully differentiable token merging using a straight-through estimator. By reducing redundant tokens in self-attention, this method improves inference speed and training efficiency without sacrificing adaptation performance.

## 4 Datasets and Protocols

PEFT methods for ViTs are commonly evaluated on a diverse set of image recognition benchmarks that vary in scale, granularity, and task complexity. While this diversity enables broad empirical assessment, both dataset characteristics and evaluation protocols can introduce biases that complicate fair and reproducible comparison across studies.

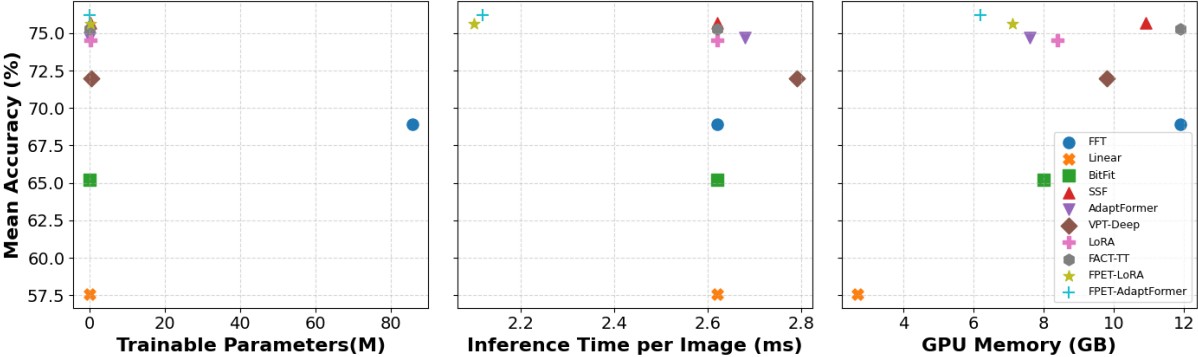

**Figure 3: Accuracy-efficiency trade-offs for selected PEFT methods applied to ViTs in image classification on VTAB-1k. The figure shows mean classification accuracy versus trainable parameters, inference time per image, and GPU memory usage.**

**Fine-grained visual classification (FGVC)** datasets, including CUB-200-2011[95], NABirds[33], Oxford Flowers[68], Stanford Dogs[44], and Stanford Cars[24], emphasize subtle inter-class differences and are frequently used to assess the adaptation capacity of PEFT methods in low-data regimes. Typical evaluation protocols involve fine-tuning on limited training splits and reporting top-1 accuracy. However, these datasets are relatively small and visually homogeneous, which can favor adaptation methods that exploit strong inductive biases while offering limited insight into scalability, robustness, or performance under distribution shift.

**VTAB-1k** provides a standardized multi-task benchmark for assessing parameter-efficient adaptation with a fixed budget of 1,000 labeled examples per task. The benchmark is organized into three categories: *Natural datasets:* CIFAR-100 [50], Caltech-101 [21], DTD [13], Flowers102 [68], Pets [71], SVHN [66], Sun397 [104]. *Specialized datasets:* Patch Camelyon [94], EuroSAT [30], Resisc45 [12], Retinopathy [26]. *Structured datasets:* CLEVR-count/distance [43], DMLab [3], KITTI [25],dSprites [63], SmallNORBs[51]. VTAB-1k enforces consistent training budgets and evaluation metrics, typically reporting mean accuracy across task groups. However, its reliance on fixed hyperparameters, short training schedules, and small data budgets can bias results toward methods optimized for low-data adaptation and may underrepresent performance differences that emerge at larger scales.

**General image recognition** datasets such as CIFAR-10[50], CIFAR-100[50], and ImageNet-1k[14] are widely used to study scalability and transfer performance. Protocols on these datasets vary substantially across studies, including differences in backbone initialization, optimizer choice, learning-rate schedules, number of training epochs, and parameter-freezing strategies. While ImageNet-1k serves as the primary large-scale benchmark for PEFT methods, inconsistent reporting of training budgets, parameter counts, and computational cost often hinders direct comparison between approaches.

**Domain generalization** benchmarks, e.g., ImageNet-V2[79], ImageNet-Sketch[97], ImageNet-A[32], and ImageNet-R[31], are used to evaluate robustness under distribution shift. These datasets are typically evaluated using zero-shot or in-distribution fine-tuned models without additional adaptation. Despite their relevance, such

benchmarks are less frequently included in PEFT evaluations, limiting current understanding of how parameter-efficient adaptation affects robustness and out-of-distribution generalization.

Across datasets such VTAB-1k and FGVC, most PEFT methods follow the evaluation protocols stated in [38]. Many studies report single-run results without variance estimates, and few normalize comparisons by computational budget or inference cost. These protocol-level inconsistencies introduce implicit biases and reduce reproducibility, highlighting the need for more standardized evaluation practices, clearer reporting of training and inference constraints, and broader inclusion of robustness-oriented benchmarks in future PEFT research.

## 5 Analysis and Discussions

### 5.1 Comparative Trade-offs

Table 1 provides a comprehensive comparison of state-of-the-art PEFT methods evaluated on VTAB-1k, grouped into natural, specialized, and structured visual tasks. Several consistent trends emerge across PEFT families, highlighting the trade-offs between accuracy, parameter efficiency, and task complexity. **Reparameterization-based** methods, including LoRA and its variants, demonstrate strong and stable performance across all VTAB categories. These approaches achieve accuracy close to or exceeding traditional full fine-tuning while updating only a small fraction of model parameters. Their effectiveness is particularly pronounced on natural image classification tasks, where low-rank updates appear sufficient to capture task-specific variations. However, performance gains on structured tasks are more modest, suggesting that purely linear reparameterization may be less expressive for tasks requiring spatial reasoning or geometric understanding. **Additive-based** approaches, such as adapters and AdaptFormer variants, offer a favorable balance between expressivity and efficiency. These methods consistently outperform selective tuning baselines and exhibit robust performance on both natural and specialized tasks. Their stronger results on structured datasets indicate that inserting lightweight task-specific modules can enhance representational flexibility beyond what is achievable through reparameterization alone. The trade-off is a moderate increase in trainable parameters and inference overhead relative to reparameterization-based methods.

| | #Param (M) | Natural | | | | | | | Specialized | | | | Structured | | | | | | | | Average |
|---|---|---|---|---|---|---|---|---|---|---|---|---|---|---|---|---|---|---|---|---|---|
| | | Cifar100 | Caltech101 | DTD | Flowers102 | Pets | SVHN | Sun397 | Camelyon | EuroSAT | Resisc45 | Retinopathy | Clevr-Count | Clevr-Dist | DMLab | KITTI-Dist | dSpr-Loc | dSpr-Ori | sNORB-Azim | sNORB-Ele | |
| *Traditional Fine-Tuning* | | | | | | | | | | | | | | | | | | | | | |
| Full | 85.8 | 68.9 | 87.7 | 64.3 | 97.2 | 86.9 | 87.4 | 38.8 | 79.7 | 95.7 | 84.2 | 73.9 | 56.3 | 58.6 | 41.7 | 65.5 | 57.5 | 46.7 | 25.7 | 29.1 | 68.9 |
| Linear Probe[49] | 0.04 | 64.4 | 85.0 | 63.2 | 97.0 | 86.3 | 36.6 | 51.0 | 78.5 | 87.5 | 68.5 | 74.0 | 34.3 | 30.6 | 33.2 | 55.4 | 12.5 | 20.0 | 9.6 | 19.2 | 57.6 |
| *PETL Methods* | | | | | | | | | | | | | | | | | | | | | |
| BitFit [112] | 0.10 | 72.8 | 87.0 | 59.2 | 97.5 | 85.3 | 59.9 | 51.4 | 78.7 | 91.6 | 72.9 | 69.8 | 61.5 | 55.6 | 32.4 | 55.9 | 66.6 | 40.0 | 15.7 | 25.1 | 65.2 |
| GPS [118] | 0.25 | 81.1 | 94.2 | 75.8 | 99.4 | 91.7 | 91.6 | 52.4 | 87.9 | 96.2 | 86.5 | 76.5 | 79.9 | 62.6 | 55.0 | 82.4 | 84.0 | 55.4 | 29.7 | 46.1 | 77.5 |
| SCT [119] | 0.11 | 75.3 | 91.6 | 72.2 | 99.2 | 91.1 | 91.2 | 55.0 | 85.0 | 96.1 | 86.3 | 76.2 | 81.1 | 65.1 | 51.7 | 80.2 | 75.4 | 46.2 | 33.2 | 45.7 | 76.0 |
| Adapter [34] | 0.16 | 69.2 | 90.1 | 68.0 | 98.8 | 89.9 | 82.8 | 54.3 | 84.0 | 94.9 | 81.9 | 75.5 | 80.9 | 65.3 | 48.6 | 78.3 | 74.8 | 48.5 | 29.9 | 41.6 | 73.9 |
| AdaptFormer [10] | 0.16 | 70.8 | 91.2 | 70.5 | 99.1 | 90.9 | 86.6 | 54.8 | 83.0 | 95.8 | 84.4 | 76.3 | 81.9 | 64.3 | 49.3 | 80.3 | 76.3 | 45.7 | 31.7 | 41.1 | 74.7 |
| Convpass [40] | 0.33 | 72.3 | 91.2 | 72.2 | 99.2 | 90.9 | 91.3 | 54.9 | 84.2 | 96.1 | 85.3 | 75.6 | 82.3 | 67.9 | 51.3 | 80.0 | 85.9 | 53.1 | 36.4 | 44.4 | 76.6 |
| Bi-AdaptFormer [42] | 0.59 | 74.1 | 92.4 | 72.1 | 99.3 | 91.6 | 89.0 | 56.3 | 88.2 | 95.2 | 86.0 | 76.2 | 83.9 | 63.9 | 53.0 | 81.4 | 86.2 | 54.8 | 35.2 | 41.3 | 77.0 |
| VPT-Shallow [38] | 0.24 | 77.7 | 86.9 | 62.6 | 97.5 | 87.3 | 74.5 | 51.2 | 78.2 | 92.0 | 75.6 | 72.9 | 50.5 | 58.6 | 40.5 | 67.1 | 68.7 | 36.1 | 20.2 | 34.1 | 67.8 |
| VPT-Deep [38] | 0.53 | 78.8 | 90.8 | 65.8 | 98.0 | 88.3 | 78.1 | 49.6 | 81.8 | 96.1 | 83.4 | 68.4 | 68.5 | 60.0 | 46.5 | 72.8 | 73.6 | 47.9 | 32.9 | 37.8 | 72.0 |
| E²VPT [27] | 0.25 | 78.6 | 89.4 | 67.8 | 98.2 | 88.5 | 85.3 | 52.3 | 82.5 | 96.8 | 84.8 | 73.6 | 71.7 | 61.2 | 47.9 | 75.8 | 80.8 | 48.1 | 31.7 | 41.9 | 73.9 |
| LoRA [35] | 0.29 | 67.1 | 91.4 | 69.4 | 98.8 | 90.4 | 85.3 | 54.0 | 84.9 | 95.3 | 84.4 | 73.6 | 82.9 | 69.2 | 48.9 | 78.5 | 75.7 | 47.1 | 31.0 | 44.0 | 74.5 |
| Bi-LoRA [42] | 1.18 | 72.1 | 91.7 | 71.2 | 99.1 | 91.4 | 90.2 | 55.8 | 87.0 | 95.4 | 85.5 | 75.5 | 83.1 | 64.1 | 52.2 | 81.2 | 86.4 | 53.5 | 36.7 | 44.4 | 76.7 |
| KARST [123] | 0.33 | 76.8 | 93.2 | 75.1 | 99.3 | 92.2 | 91.9 | 57.6 | 88.3 | 96.2 | 88.4 | 75.7 | 83.8 | 69.0 | 52.9 | 82.0 | 86.0 | 52.9 | 33.8 | 47.0 | 78.1 |
| FacT-TT [41] | 0.04 | 71.3 | 89.6 | 70.7 | 98.9 | 91.0 | 87.8 | 54.6 | 82.0 | 95.5 | 83.4 | 75.7 | 82.0 | 69.0 | 49.8 | 80.0 | 79.2 | 48.4 | 34.2 | 41.4 | 75.3 |
| FacT-TK [41] | 0.01 | 70.6 | 90.6 | 70.8 | 99.1 | 90.7 | 88.6 | 54.1 | 84.8 | 96.2 | 84.5 | 75.7 | 82.6 | 68.2 | 49.8 | 80.7 | 80.8 | 47.4 | 33.2 | 43.0 | 75.6 |
| OFT [76] | 0.15 | 68.8 | 91.9 | 73.8 | 99.7 | 92.2 | 91.8 | 49.2 | 90.2 | 100 | 89.1 | 80.5 | 83.2 | 71.1 | 53.9 | 81.3 | 82.0 | 54.3 | 34.4 | 43.8 | 78.0 |
| GOFT [60] | 0.02 | 75.0 | 93.9 | 72.3 | 99.7 | 92.6 | 85.2 | 60.9 | 89.1 | 100 | 87.9 | 82.4 | 84.0 | 74.2 | 55.1 | 82.0 | 80.9 | 52.7 | 32.3 | 43.8 | 78.6 |
| Hydra [47] | 0.28 | 72.7 | 91.3 | 72.0 | 99.2 | 91.4 | 90.7 | 55.5 | 85.8 | 96.0 | 86.1 | 75.9 | 83.2 | 68.2 | 50.9 | 82.3 | 80.3 | 50.8 | 34.5 | 43.1 | 76.5 |
| SSF [54] | 0.24 | 69.0 | 92.6 | 75.1 | 99.4 | 91.8 | 90.2 | 52.9 | 87.4 | 95.9 | 87.4 | 75.5 | 75.9 | 62.3 | 53.3 | 80.6 | 77.3 | 54.9 | 29.5 | 37.9 | 75.7 |
| NOAH [117] | 0.36 | 69.6 | 92.7 | 70.2 | 99.1 | 90.4 | 86.1 | 53.7 | 84.4 | 95.4 | 83.9 | 75.8 | 82.8 | 68.9 | 49.9 | 81.7 | 81.8 | 48.3 | 32.8 | 44.2 | 75.5 |
| RepAdapter [59] | 0.22 | 69.0 | 92.6 | 75.1 | 99.4 | 91.8 | 90.2 | 52.9 | 87.4 | 95.9 | 87.4 | 75.5 | 75.9 | 62.3 | 53.3 | 80.6 | 77.3 | 54.9 | 29.5 | 37.9 | 76.1 |
| LoSA [65] | 0.19 | 82.5 | 92.8 | 76.1 | 99.7 | 90.5 | 82.0 | 55.8 | 86.6 | 97.1 | 87.0 | 76.7 | 81.5 | 62.3 | 48.6 | 82.1 | 94.2 | 61.7 | 47.9 | 45.6 | 78.4 |
| DTL [22] | 0.05 | 74.1 | 94.8 | 71.8 | 99.4 | 91.7 | 90.4 | 57.2 | 87.9 | 96.7 | 87.5 | 74.8 | 81.9 | 64.7 | 51.5 | 81.9 | 93.9 | 54.0 | 35.6 | 50.3 | 76.7 |
| LAST [88] | 0.66 | 66.7 | 93.4 | 76.1 | 99.6 | 89.8 | 86.1 | 54.3 | 86.2 | 96.3 | 86.8 | 75.4 | 81.9 | 65.9 | 49.4 | 82.6 | 87.9 | 46.7 | 32.3 | 51.5 | 76.5 |
| DyT [120] | 0.16 | 74.4 | 95.5 | 73.6 | 99.2 | 91.7 | 87.5 | 57.4 | 88.3 | 96.1 | 86.7 | 76.7 | 83.5 | 63.5 | 52.9 | 83.1 | 85.7 | 54.9 | 34.3 | 45.9 | 77.6 |
| FPET-RepAdapter [45] | 0.23 | 72.1 | 91.5 | 71.8 | 99.3 | 90.7 | 90.3 | 55.0 | 85.2 | 96.2 | 84.5 | 75.6 | 82.2 | 67.7 | 49.7 | 79.9 | 82.2 | 48.7 | 36.9 | 41.7 | 76.1 |
| FPET-LoRA [45] | 0.30 | 70.1 | 92.7 | 69.4 | 99.1 | 90.8 | 85.4 | 55.6 | 87.2 | 94.6 | 82.5 | 74.1 | 83.0 | 63.4 | 50.6 | 81.6 | 84.7 | 51.5 | 34.3 | 43.3 | 75.6 |
| FPET-AdaptFormer [45] | 0.17 | 71.3 | 93.5 | 69.9 | 99.3 | 90.7 | 87.0 | 54.7 | 87.5 | 95.1 | 84.5 | 76.2 | 83.6 | 63.1 | 52.2 | 81.3 | 87.1 | 54.1 | 33.5 | 40.2 | 76.2 |
| FPET-Bi-LoRA [45] | 1.18 | 71.9 | 91.1 | 70.9 | 99.1 | 90.5 | 89.4 | 55.9 | 87.4 | 94.7 | 84.4 | 74.9 | 83.5 | 65.1 | 52.1 | 79.7 | 85.8 | 54.2 | 36.7 | 44.4 | 76.4 |
| FPET-Bi-AdaptFormer [45] | 0.64 | 74.1 | 92.8 | 72.5 | 99.4 | 91.1 | 89.6 | 56.2 | 88.3 | 94.9 | 86.3 | 75.3 | 83.8 | 63.0 | 52.8 | 81.4 | 85.7 | 54.4 | 35.9 | 42.2 | 77.0 |

**Table 1: VTAB-1k results for representative PEFT methods, reporting mean accuracy (Average) across natural, specialized, and structured task groups. All methods use ViT-B/16 pretrained on ImageNet-21K as the backbone.**

**Selective tuning** methods, including BitFit and linear probing, achieve the lowest parameter cost but also exhibit the largest performance gap relative to other PEFT families. While these approaches remain competitive on some natural image tasks, their limited expressive capacity becomes evident on specialized and structured datasets, where more substantial adaptation is required. This highlights an inherent trade-off between extreme parameter efficiency and task generalization. **Hybrid-based** methods, which combine complementary strategies such as reparameterization and additive adaptation, consistently achieve strong performance across all VTAB groups. In particular, hybrid approaches demonstrate improved robustness on structured tasks, narrowing the gap with full fine-tuning while maintaining high parameter efficiency. These results suggest that combining multiple adaptation mechanisms can mitigate the limitations of individual PEFT paradigms. Finally, **inference-efficient** variants, including FPET-based methods, further improve deployment efficiency without sacrificing accuracy. These approaches retain competitive performance across VTAB categories while reducing inference cost, making them attractive for resource-constrained and large-scale deployment scenarios. Overall, the results in Table 1 confirm that no single PEFT strategy universally dominates across all tasks. Instead, different PEFT families occupy distinct regions of the accuracy–efficiency trade-off space, with performance strongly influenced by dataset characteristics and task complexity. These findings reinforce the importance of selecting PEFT methods based on application-specific constraints rather than accuracy alone.

Under the evaluation protocols of [38, 45], Figure 3 provides a comparative view of how selected PEFT methods balance predictive performance against computational and memory costs. The left panel highlights the substantial parameter overhead of full fine-tuning, which achieves competitive accuracy but requires updating nearly the entire model. In contrast, PEFT methods attain comparable or improved accuracy while updating only a small fraction of parameters, demonstrating the effectiveness of parameter-efficient adaptation strategies. Reparameterization-based approaches, such as LoRA and its FPET variants, consistently achieve strong accuracy with minimal trainable parameters, indicating a favorable accuracy-efficiency trade-off. Notably, FPET-LoRA and FPET-AdaptFormer achieve the highest accuracy among the considered methods while maintaining low parameter counts, suggesting that combining low-rank reparameterization with structured fine-tuning can further enhance representational capacity. Additive-based methods such as AdaptFormer also perform competitively, though with slightly higher inference cost compared to purely reparameterization-based techniques. The middle panel reveals that most PEFT methods incur only modest inference-time overhead relative to full fine-tuning, with differences largely attributable to additional adapter modules. Importantly, methods with higher accuracy do not necessarily require increased inference time, underscoring that parameter efficiency and inference efficiency are not strictly opposing objectives. Selective tuning methods, exemplified by BitFit, offer minimal overhead but exhibit a noticeable drop in accuracy, reflecting their limited expressive capacity. The right panel further emphasizes

memory efficiency, showing that PEFT methods substantially reduce GPU memory usage compared to full fine-tuning. FPET-based approaches, occupy a favorable region of the accuracy-memory trade-off space, making them particularly attractive for deployment in resource-constrained or large-scale settings.

## 5.2 Strengths and Weaknesses

**Strengths.** PEFTs has emerged as a powerful paradigm for adapting large ViTs to downstream image classification tasks while significantly reducing training cost. One of its primary strengths lies in its ability to achieve competitive accuracy with only a small fraction of trainable parameters. By freezing most of the pretrained backbone, PEFT methods substantially lower memory consumption, training time, and energy usage, making them well suited for scenarios involving limited computational resources or large numbers of downstream tasks. Another key advantage of PEFT is improved scalability across tasks. Since task-specific parameters are lightweight, multiple adaptations can be stored and deployed efficiently, enabling practical multi-task and continual learning setups. In addition, the constrained parameter updates imposed by PEFT often act as an implicit regularizer, which can mitigate overfitting in low-data regimes and lead to improved generalization compared to full fine-tuning. This effect is particularly evident in reparameterization-based and hybrid PEFT approaches, which balance expressive capacity with strong inductive biases.

**Weaknesses.** Despite these advantages, PEFT methods also exhibit notable limitations. A central weakness is their sensitivity to design choices, such as insertion points[69], rank selection[23, 62], or prompt length[38, 101], which can significantly affect performance. Unlike full fine-tuning, PEFT does not universally guarantee strong adaptation across all tasks and dataset scales, and poorly chosen configurations may underperform even simple baselines. Inference efficiency represents another important challenge. While PEFT methods reduce training cost, some approaches introduce additional computational overhead at inference time due to extra modules[10, 34]. As a result, parameter efficiency does not always translate directly into inference efficiency, particularly for deployment on latency-critical systems.

PEFT presents a compelling trade-off space rather than a universally optimal solution. Its strengths in efficiency, scalability, and generalization must be weighed against limitations in expressiveness, inference overhead, and sensitivity to hyperparameter choices. These observations highlight the need for principled design guidelines, unified benchmarks, and inference-aware PEFT strategies, which we identify as important directions for future research.

## 5.3 Future Directions

Although PEFT has demonstrated strong empirical performance for adapting ViTs, several open research challenges remain. Addressing these challenges is essential for improving the scalability, robustness, and theoretical understanding of PEFT in real-world deployment settings.

**Structure-aware adaptation.** Most existing PEFT methods treat tokens independently during adaptation, ignoring structural relationships such as spatial locality or semantic similarity. Future work could explore structure-aware PEFT strategies that leverage

token graphs or region-level relations to guide parameter updates. Such approaches may reduce redundant adaptations and improve generalization, particularly for complex visual scenes.

**Dynamic and input-adaptive PEFT.** Current PEFT methods rely on static parameter allocation, applying the same adaptation modules to all inputs regardless of difficulty or content. A promising direction is dynamic PEFT, where adapters or low-rank components are conditionally activated based on input characteristics, token importance, or task uncertainty. This could enable more efficient computation and better accuracy-efficiency trade-offs, which are critical for web-scale and real-time applications.

**Robustness and distribution shift.** PEFT models are typically evaluated under clean, in-distribution settings, leaving their robustness underexplored. Future work should systematically study PEFT behavior under distribution shifts, data scarcity, and adversarial perturbations. Incorporating robustness-aware objectives or regularization strategies into PEFT could improve reliability in practical, open-world environments.

**Theoretical foundations.** Despite their empirical success, PEFTs lack rigorous theoretical explanations. Open questions remain regarding the expressivity and generalization properties of low-rank updates, adapters, and partial fine-tuning. Developing theoretical analyses of optimization dynamics and representation adaptation would provide principled guidance for future PEFT designs.

**Deployment considerations.** Finally, the field would benefit from standardized benchmarks that evaluate PEFT methods across accuracy, parameter efficiency, computational cost, and robustness. Practical deployment constraints, including memory footprint and inference latency, should be explicitly considered to bridge the gap between research and real-world applications.

## 6 Conclusion

PEFT has emerged as an effective paradigm for adapting large ViTs to image classification tasks under practical computational constraints. Updating only a small subset of parameters substantially reduce training cost, memory usage, and energy consumption while maintaining competitive predictive performance.

In this survey, we presented a structured taxonomy and comparative analysis of PEFT approaches. Our analysis highlights the effectiveness of localized and low-dimensional adaptations for capturing task-specific information. While early PEFTs focused on static and layer-wise efficiency, recent work has increasingly explored inference-efficient strategies that improve accuracy-efficiency trade-offs. Despite their promise, several challenges remain. Current evaluations largely focus on clean, in-distribution benchmarks, leaving robustness, distribution shift, and low-data performance underexplored. In addition, the theoretical understanding of PEFT regarding expressivity and generalization remains limited. Addressing these gaps will be crucial for advancing PEFT toward more principled, robust, and widely deployable image classification systems.

## Acknowledgments

Edwin Kwadwo Tenagyei is supported by the Griffith University International Postgraduate Research Scholarship and the Griffith University Postgraduate Research Scholarship. This research was conducted under the supervision of Lei Wang.

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

## A  Comparison with Existing Surveys

Parameter-efficient fine-tuning (PEFT) has emerged as a practical alternative to full fine-tuning for adapting large pretrained models, particularly transformer models, to downstream tasks under constraints on data, computation, and storage. This paradigm has motivated a growing literature of surveys and reviews that aim to systematize efficient adaptation techniques, typically grouping methods such as adapters, prompt-based tuning, low-rank parameterizations, bias or normalization tuning, and selective fine-tuning under a unified framework. Many of these surveys earlier placed strong emphasis on natural language processing and large language models [126, 128, 136, 138].

Within the vision domain, several studies [137, 139] survey PEFT strategies for Vision Transformers (ViTs), while a parallel line of work adopts a cross-modality perspective, framing vision adaptation alongside PEFT methods developed for language and multimodal architectures [4, 73, 142]. These studies provide valuable overviews of early design choices, common architectural insertion points, and representative empirical results on benchmarks such as VTAB-1k, ImageNet variants, and fine-grained visual classification (FGVC) datasets. However, as the PEFT landscape for ViTs has rapidly expanded, existing surveys often provide limited coverage of more recent and hybrid approaches, specifically methods that explicitly exploit the token-based structure of ViTs for better inference efficiency of PEFTs. Moreover, prior surveys typically emphasize categorical taxonomies of methods, but devote less attention to comparative trade-offs across PEFT families when evaluated under consistent constraints. In particular, distinctions between performance gains arising from increased trainable capacity, improved inductive bias, or task-specific adaptation are not always made explicit. Similarly, differences in PEFT behavior across task regimes such as natural image classification versus specialized domains or structured reasoning tasks are often underexplored, despite their importance for understanding generalization and deployment. These developments raise new questions about scalability and stability that extend beyond traditional notions of parameter efficiency and motivate a more architecture-grounded synthesis.

Different from other survey, this survey focuses specifically on PEFTs for ViTs in image classification, organizing existing methods and emphasizing empirical and practical trade-offs across PEFT families. By consolidating both established and emerging approaches within a unified, vision-centric perspective, the survey complements existing PEFT reviews while addressing gaps in coverage, comparison, and architectural specificity.

## B  Benchmark Dataset Statistics

Table 2 shows the overall statistics of benchmark datasets.

## C  Other Benchmark Evaluations

### C.1  Fine-Grained Visual Classification

Table 3 presents a comparative evaluation of representative PEFT methods on FGVC benchmarks, including CUB-200-2011, NABirds, Oxford Flowers, Stanford Dogs, and Stanford Cars, using a ViT-B/16 backbone pretrained on ImageNet-21K. These datasets are characterized by subtle inter-class differences and high intra-class similarity, making them particularly challenging and well suited for assessing the expressive capacity of PEFT methods under limited adaptation budgets. Several observations emerge from the results. First, PEFT methods consistently outperform linear probing and closely match full fine-tuning performance while using orders of magnitude fewer trainable parameters. This highlights the effectiveness of parameter-efficient adaptation in capturing fine-grained visual cues without requiring full model updates.

Among PEFT families, reparameterization-based methods, such as LoRA, demonstrate strong and stable performance across FGVC datasets, achieving competitive mean accuracy with relatively modest parameter counts. This suggests that low-rank updates to attention weights are sufficient to adapt pretrained representations to fine-grained classification tasks. Additive-based approaches, including adapters and RepAdapter, further improve performance on several datasets, particularly CUB-200 and Cars, indicating that inserting lightweight task-specific modules can enhance representational flexibility for distinguishing subtle visual attributes. In contrast, prompt-based methods exhibit more mixed behavior. While VPT-Deep outperforms its shallow counterpart and improves over linear probing, its performance generally lags behind adapter and LoRA-based approaches, especially on datasets requiring detailed part-level discrimination. This suggests that prompt tuning alone may offer limited expressivity for fine-grained tasks. Importantly, the results also reveal clear accuracy-parameter trade-offs. Methods such as RepAdapter achieve the highest mean accuracy among PEFT approaches while maintaining a relatively small parameter footprint, whereas more parameter-constrained methods, such as BitFit, offer lower performance but minimal adaptation cost. These trade-offs underscore that no single PEFT strategy is universally optimal but instead, the choice of method should be guided by task difficulty and deployment constraints.

The FGVC results confirm that PEFT methods can effectively adapt large ViTs to fine-grained recognition tasks with minimal parameter updates. They further highlight the advantages of additive and reparameterization-based adaptations over purely selective or shallow prompt-based approaches in settings that demand high representational precision.

### C.2  Evaluation on Hierarchical Transformers

Table 4 compares full fine-tuning, linear probing, and a range of PEFT methods on VTAB-1k using a Swin-pretrained model. Several important trends emerge.

First, full fine-tuning achieves strong overall performance (75.0% average accuracy), particularly on Natural tasks (79.2%), but at a very high parameter cost (86.7M trainable parameters). This highlights the effectiveness of end-to-end adaptation, but also underscores its limited practicality in low-data or resource-constrained settings. Second, linear probing performs substantially worse, especially on Structured tasks (33.5%), confirming that freezing the backbone severely restricts adaptability to distributional shifts and non-natural image structures. This gap demonstrates that some form of feature adaptation is essential for VTAB-style transfer. Third, PEFT methods consistently outperform linear probing and approach, or surpass, full fine-tuning while updating fewer than 0.5M parameters. Among them, Task-adaptive methods such as FacT, DTL, and KARST show the strongest performance, with KARST achieving

**Table 2: Common datasets in the visual PEFT domain for image recognition tasks. For each dataset, we report the number of classes and the sizes of training, validation, and test splits.**

| Dataset | Description | #Classes | Train | Val | Test |
|---|---|---|---|---|---|
| **Fine-Grained Visual Classification (FGVC)** | | | | | |
| CUB-200-2011 | Fine-grained bird species recognition | 200 | 5,394 | 600 | 5,794 |
| NABirds | Fine-grained bird species recognition | 55 | 21,536 | 2,393 | 24,633 |
| Oxford Flowers | Fine-grained flower species recognition | 102 | 1,020 | 1,020 | 6,149 |
| Stanford Dogs | Fine-grained dog species recognition | 120 | 10,800 | 1,200 | 8,580 |
| Stanford Cars | Fine-grained car classification | 196 | 7,329 | 815 | 8,041 |
| **Visual Task Adaptation Benchmark (VTAB-1k) [141]** | | | | | |
| CIFAR-100 | Natural image classification | 100 | – | – | 10,000 |
| Caltech101 | Natural image classification | 102 | – | – | 6,084 |
| DTD | Texture recognition | 47 | – | – | 1,880 |
| Flowers102 | Natural images captured using standard cameras | 102 | 800 | 200 | 6,149 |
| Pets | Natural image classification | 37 | – | – | 3,669 |
| SVHN | Street view house numbers | 10 | – | – | 26,032 |
| Sun397 | Scene recognition | 397 | – | – | 21,750 |
| Patch Camelyon | Specialized medical image classification | 2 | – | – | 32,768 |
| EuroSAT | Remote sensing image classification | 10 | 800 | 200 | 5,400 |
| Resisc45 | Remote sensing scene classification | 45 | – | – | 6,300 |
| Retinopathy | Medical image diagnosis | 5 | – | – | 42,670 |
| Clevr/count | Structured geometric reasoning (counting) | 8 | – | – | 15,000 |
| Clevr/distance | Structured geometric reasoning (distance) | 6 | – | – | 15,000 |
| DMLab | 3D visual reasoning | 6 | – | – | 22,735 |
| KITTI/distance | Depth and distance estimation | 4 | 800 | 200 | 711 |
| dSprites/location | Disentangled representation learning | 16 | – | – | 73,728 |
| dSprites/orientation | Disentangled representation learning | 16 | – | – | 73,728 |
| SmallNORB/azimuth | 3D object recognition | 18 | – | – | 12,150 |
| SmallNORB/elevation | 3D object recognition | 9 | – | – | 12,150 |
| **General Image Recognition Datasets** | | | | | |
| CIFAR-10 | General image recognition | 10 | 50,000 | – | 10,000 |
| CIFAR-100 | General image recognition | 100 | 50,000 | – | 10,000 |
| ImageNet-1k | Large-scale image classification | 1,000 | 1,281,167 | 50,000 | 50,000 |
| **Domain Generalization Datasets** | | | | | |
| ImageNet-V2 | Domain generalization recognition | 1,000 | – | – | 10,000 |
| ImageNet-Sketch | Domain generalization recognition | 1,000 | – | – | 50,889 |
| ImageNet-A | Adversarial natural images | 200 | – | – | 7,500 |
| ImageNet-R | Artistic renditions | 200 | – | – | 30,000 |

| Method | Params.(M) | Average | Natural | Specialized | Structured |
|---|---|---|---|---|---|
| Full fine-tuning | 86.7 | 75.0 | 79.2 | 86.2 | 59.7 |
| Linear probing[49] | 0 | 62.6 | 73.5 | 80.8 | 33.5 |
| Bitfit[112] | 0.20 | 65.6 | 74.2 | 80.1 | 42.4 |
| VPT-Shallow [38] | 0.00 | 66.7 | 79.9 | 82.5 | 37.8 |
| VPT-Deep [38] | 0.16 | 71.6 | 76.8 | 84.5 | 53.4 |
| FacT[38] | 0.14 | 77.4 | 83.1 | 86.9 | 62.1 |
| DTL[22] | 0.09 | 77.9 | 82.4 | 87.0 | 64.2 |
| KARST[123] | 0.45 | 78.6 | 83.9 | 87.7 | 64.2 |

**Table 4: Results on VTAB-1k with Swin-B as backbone. Params.(M): number of trainable parameters in backbones.**

| Method | CUB-200 | NABirds | Flowers | Dogs | Cars | Mean | Params.(M) |
|---|---|---|---|---|---|---|---|
| Full fine-tuning | 87.3 | 82.7 | 98.8 | 89.4 | 84.5 | 88.5 | 85.98 |
| Linear probing[49] | 85.3 | 75.9 | 97.9 | 86.2 | 51.3 | 79.3 | 0.18 |
| Bitfit[112] | 88.4 | 84.2 | 98.8 | 91.2 | 79.4 | 88.4 | 0.28 |
| VPT-Shallow [38] | 86.7 | 78.8 | 98.4 | 90.7 | 68.7 | 84.7 | 0.25 |
| VPT-Deep [38] | 88.5 | 84.2 | 99.0 | 90.2 | 83.6 | 89.1 | 0.85 |
| SSF [54] | 82.7 | 85.9 | 98.5 | 87.7 | 82.6 | 87.5 | 0.39 |
| LoRA [35] | 88.3 | 85.6 | 99.2 | 91.0 | 83.2 | 89.5 | 0.44 |
| Adapter [34] | 87.1 | 84.3 | 98.5 | 89.8 | 68.6 | 85.7 | 0.41 |
| RepAdapter [59] | 89.4 | 86.5 | 99.5 | 90.6 | 85.9 | 90.3 | 0.34 |

**Table 3: Performance comparison of representative PEFT methods on FGVC datasets, including CUB-200-2011, NABirds, Oxford Flowers, Stanford Dogs, and Stanford Cars. Results are reported as top-1 accuracy (%). All methods use ViT-B/16 pretrained on ImageNet-21K as the backbone.**

the highest average accuracy (78.6%) and the best results across all three VTAB subsets. Notably, KARST improves Structured task accuracy by over 30 points relative to linear probing, indicating superior inductive bias for tasks involving geometric or symbolic structure. Fourth, Structured tasks remain the most challenging subset, but also the one where PEFT methods deliver the largest relative gains. This suggests that parameter-efficient adaptations are particularly effective at injecting task-specific inductive biases into hierarchical ViTs like Swin, without overfitting the limited VTAB-1k supervision.

These results demonstrate that carefully designed PEFT strategies can not only match but exceed full fine-tuning on VTAB-1k, achieving a better performance-efficiency trade-off.

## References for Appendix

[4] Jieming Bian, Yuanzhe Peng, Lei Wang, Yin Huang, and Jie Xu. 2025. A survey on parameter-efficient fine-tuning for foundation models in federated learning.

*arXiv preprint arXiv:2504.21099* (2025).

[126] Ning Ding, Yujia Qin, Guang Yang, Fuchao Wei, Zonghan Yang, Yusheng Su, Shengding Hu, Yulin Chen, Chi-Min Chan, Weize Chen, et al. 2022. Delta tuning: A comprehensive study of parameter efficient methods for pre-trained language models. *arXiv preprint arXiv:2203.06904* (2022).

[22] Minghao Fu, Ke Zhu, and Jianxin Wu. 2024. Dtl: Disentangled transfer learning for visual recognition. In *Proceedings of the AAAI conference on artificial intelligence*, Vol. 38. 12082–12090.

[128] Zeyu Han, Chao Gao, Jinyang Liu, Jeff Zhang, and Sai Qian Zhang. 2024. Parameter-efficient fine-tuning for large models: A comprehensive survey. *arXiv preprint arXiv:2403.14608* (2024).

[34] Neil Houlsby, Andrei Giurgiu, Stanislaw Jastrzebski, Bruna Morrone, Quentin De Laroussilhe, Andrea Gesmundo, Mona Attariyan, and Sylvain Gelly. 2019. Parameter-efficient transfer learning for NLP. In *International conference on machine learning*. PMLR, 2790–2799.

[35] Edward J Hu, Yelong Shen, Phillip Wallis, Zeyuan Allen-Zhu, Yuanzhi Li, Shean Wang, Lu Wang, Weizhu Chen, et al. 2022. Lora: Low-rank adaptation of large language models. *ICLR* 1, 2 (2022), 3.

[38] Menglin Jia, Luming Tang, Bor-Chun Chen, Claire Cardie, Serge Belongie, Bharath Hariharan, and Ser-Nam Lim. 2022. Visual prompt tuning. In *European conference on computer vision*. Springer, 709–727.

[49] Simon Kornblith, Jonathon Shlens, and Quoc V Le. 2019. Do better imagenet models transfer better?. In *Proceedings of the IEEE/CVF conference on computer vision and pattern recognition*. 2661–2671.

[54] Dongze Lian, Daquan Zhou, Jiashi Feng, and Xinchao Wang. 2022. Scaling & shifting your features: A new baseline for efficient model tuning. *Advances in Neural Information Processing Systems* 35 (2022), 109–123.

[59] Gen Luo, Minglang Huang, Yiyi Zhou, Xiaoshuai Sun, Guannan Jiang, Zhiyu Wang, and Rongrong Ji. 2023. Towards efficient visual adaption via structural re-parameterization. *arXiv preprint arXiv:2302.08106* (2023).

[73] Nusrat Jahan Prottasha, Upama Roy Chowdhury, Shetu Mohanto, Tasfia Nuzhat, Abdullah As Sami, Md Shamol Ali, Md Shohanur Islam Sobuj, Hafijur Raman, Md Kowsher, and Ozlem Ozmen Garibay. 2025. PEFT A2Z: Parameter-Efficient Fine-Tuning Survey for Large Language and Vision Models. *arXiv preprint arXiv:2504.14117* (2025).

[136] Luping Wang, Sheng Chen, Linnan Jiang, Shu Pan, Runze Cai, Sen Yang, and Fei Yang. 2024. Parameter-efficient fine-tuning in large models: A survey of methodologies. *arXiv preprint arXiv:2410.19878* (2024).

[137] Yi Xin, Siqi Luo, Haodi Zhou, Junlong Du, Xiaohong Liu, Yue Fan, Qing Li, and Yuntao Du. 2024. Parameter-efficient fine-tuning for pre-trained vision models: A survey. *arXiv e-prints* (2024), arXiv–2402.

[138] Lingling Xu, Haoran Xie, Si-Zhao Joe Qin, Xiaohui Tao, and Fu Lee Wang. 2023. Parameter-efficient fine-tuning methods for pretrained language models: A critical review and assessment. *arXiv preprint arXiv:2312.12148* (2023).

[139] Bruce XB Yu, Jianlong Chang, Haixin Wang, Lingbo Liu, Shijie Wang, Zhiyu Wang, Junfan Lin, Lingxi Xie, Haojie Li, Zhouchen Lin, et al. 2024. Visual tuning. *Comput. Surveys* 56, 12 (2024), 1–38.

[112] Elad Ben Zaken, Yoav Goldberg, and Shauli Ravfogel. 2022. Bitfit: Simple parameter-efficient fine-tuning for transformer-based masked language-models. In *Proceedings of the 60th Annual Meeting of the Association for Computational Linguistics (Volume 2: Short Papers)*. 1–9.

[141] Xiaohua Zhai, Joan Puigcerver, Alexander Kolesnikov, Pierre Ruyssen, Carlos Riquelme, Mario Lucic, Josip Djolonga, André Susano Pinto, Maxim Neumann, Alexey Dosovitskiy, Lucas Beyer, Olivier Bachem, Michael Tschannen, Marcin Michalski, Olivier Bousquet, Sylvain Gelly, and Neil Houlsby. 2019. A Large-scale Study of Representation Learning with the Visual Task Adaptation Benchmark. *arXiv: Computer Vision and Pattern Recognition* (2019).

[142] Cheng Zhihao, Shufen Zhihao, Ming Li, Wang Jiahao, Zhang Yifan, Liu Xinyi, Chen Wei, and Sun Qian. 2025. A Comprehensive Survey of Parameter-Efficient Fine-Tuning for Large Language and Vision Models. *Authorea Preprints* (2025).

[123] Yue Zhu, Haiwen Diao, Shang Gao, Long Chen, and Huchuan Lu. 2025. KARST: Multi-Kernel Kronecker Adaptation with Re-Scaling Transmission for Visual Classification. *ICASSP 2025 - 2025 IEEE International Conference on Acoustics, Speech and Signal Processing (ICASSP)* (2025), 1–5.