# OpenReview forum: "Beyond Fine-Tuning: The Present and Future of Parameter-Efficient Fine-Tuning in Vision Transformers"
_ACM.org/TheWebConf/2026/Workshop/TIME — TIME 2026 Oral_

### Official Review · Reviewer_SvWK · 2025-12-29
**Beyond Fine-Tuning: The Present and Future of Parameter-Efficient Fine-Tuning in Vision Transformers**

**Rating:** 7
**Confidence:** 4

**Review:**

1) Move beyond criticizing inconsistent protocols  by proposing a concrete "gold standard" evaluation framework for future research.
2) Include latency metrics on edge devices (not just GPUs) to substantialize "deployment efficiency" claims.
3) Expand the scope beyond image classification to dense prediction tasks (segmentation/detection) to demonstrate broader PEFT versatility.
4) Link empirical efficiency to theoretical frameworks, such as intrinsic dimensionality, to explain why low-rank adaptations succeed.
5) Add heatmaps showing method sensitivity to key hyperparameters (e.g., rank $r$, prompt length) to quantify robustness.
6) Accompany the survey with a unified, reproducible code repository to address the reproducibility gaps identified.
7) Provide concrete guidelines or heuristics for selecting compatible components in "Hybrid-based Tuning".
8) Include a small case study or preliminary experiments on distribution shifts rather than just listing benchmarks.
9) Visually distinguish between training-efficient and inference-efficient methods in the taxonomy figure.
10) Discuss PEFT applicability to hierarchical Transformers (e.g., Swin) more explicitly, as most analysis currently defaults to standard ViT.

---

### Official Review · Reviewer_JfPy · 2025-12-31
**Beyond Fine-Tuning: The Present and Future of Parameter-Efficient Fine-Tuning in Vision Transformers**

**Rating:** 6
**Confidence:** 4

**Review:**

The authors created a great taxonomy to group different Parameter-Efficient Fine-Tuning methods. The paper explains the trade-offs between accuracy, the number of trainable parameters, and inference speed. The authors show a  growth trend of how much people are studying this topic.
A few Points to Improve
• The paper needs to test robustness more when the data has a distribution shift.
• There is not enough theoretical foundation to explain why low-rank updates work so well.
• The authors should suggest standardized evaluation protocols so all scientists can compare results fairly.
• The research should look at structure-aware adaptation to help the model understand the spatial locality of tokens.
• The work focuses too much on image classification and needs to include other tasks.
• Some methods like adapters still have latency during inference, which needs a better fix.
• The authors should talk more about how these models perform with limited labeled data

---

### Official Review · Reviewer_BxcZ · 2026-01-05
**Review of paper15**

**Rating:** 7
**Confidence:** 4

**Review:**

The authors present a systematic survey of Parameter-Efficient Fine-Tuning (PEFT) techniques tailored for Vision Transformers. By introducing a robust taxonomy—spanning additive, reparameterization, selective, and hybrid paradigms—the manuscript successfully integrates a fragmented body of literature into a cohesive framework.

A notable strength of this work is the rigorous evaluation of benchmark results, which moves beyond simple accuracy metrics to address critical trade-offs in scalability and hardware efficiency. This deployment-centric perspective is highly relevant for real-world applications. To further strengthen the manuscript, I suggest providing additional metadata for the comparisons in Figure 3. Clarifying the task-averaging logic for the VTAB-1k results and the specific benchmarking environment (e.g., hardware specifications and batch sizes) would ensure the reported efficiency gains are fully interpretable.

In summary, this is a well-executed review that provides a valuable service to the vision-learning community.

---

### Official Review · Reviewer_8neG · 2026-01-08

**Rating:** 7
**Confidence:** 4

**Review:**

Paper strength:
1)Acronyms first occurances all contain expansion.
2)Table 1 facts verified to be correct for LoRA, RepAdapter adn VPT-Deep(data cross section verified).
3)Table 3 and Fig 3 cross section fact checks confirm data accuracy.
4)Benchmarking of results in Table 1 is a major contribution of this paper.
5)Figure 3 conforms to 'Multi-facet' aspect of requirements in conference submission.
6)Survey range extends to recent papers
7)Explains limitations explicitly addressing data contraints to clean and in-distribution benchmark(conclusion section)
8)Identification of OOD is pivotal and cites Hendrycks et al as source

Weakness
1)OOD metrics would be nice to have in Table 1 or in-text.
2)Does not explain threshold ceilings with reference to 'Robustness' such as LoRA 0.29M Param and its relationship with robustness.
3)Paper seems somewhat derivative of Xin et al. 2024 paper but expands upon ViT analysis
4)Token merging concept explained in shallow way and leaves narrative incomplete

---

### Author Rebuttal · Authors · 2026-01-12

We sincerely thank all reviewers for their careful reading, positive evaluations, and constructive feedback. We are encouraged by the consistent recognition of the manuscript’s contributions, including the proposed taxonomy for Parameter-Efficient Fine-Tuning (PEFT), the rigor and accuracy of the benchmarking synthesis, the deployment-centric perspective on efficiency trade-offs, and the coverage of recent literature. The reviews collectively confirm the paper’s relevance and value to the vision-learning community.

Several comments raised important topics such as robustness under distribution shift, theoretical explanations of low-rank adaptation, standardized evaluation protocols, broader task coverage, and deployment on edge devices. We clarify that the manuscript is intentionally scoped to PEFT methods for image classification with Vision Transformers under clean, in-distribution benchmarks, which remain the most consistently reported and comparable setting across existing studies. Topics such as OOD robustness, dense prediction tasks, unified re-implementations, and new empirical case studies are therefore treated as future research directions, rather than omissions, to preserve methodological consistency in a survey setting.

Within this scope, we will make the following concrete revisions in the revised manuscript:

- Benchmarking clarity : We will clarify the task-averaging protocol used for VTAB-1k and explicitly state the benchmarking context for Figure 3 such as batch sizes as reported in the original works.
- Efficiency and latency interpretation: We will expand the discussion of accuracy–efficiency–latency trade-offs across PEFT families.
- We will expand the discussion on PEFT applicability to hierarchical Transformer architectures (e.g., Swin) in the Appendix, clarifying where insights generalize beyond standard ViT backbones.

We believe these revisions will significantly improve clarity, transparency, and practical usefulness, while preserving the focused scope and methodological consistency of the survey. We thank the reviewers again for their constructive and forward-looking feedback and believe the revised manuscript will be an even stronger resource for the community.

---

### Meta-Review · Area_Chair_Pby2 · 2026-01-16

**Recommendation:** Accept (Oral)
**Confidence:** 5

**Metareview:**

This paper presents a structured survey of parameter-efficient fine-tuning (PEFT) methods for Vision Transformers, with a clear taxonomy and a synthesis of benchmarking results focused on image classification. Reviewers consistently acknowledge the paper’s strengths, including the well-organized taxonomy, accurate and up-to-date literature coverage, careful benchmarking synthesis, and a deployment-oriented discussion of efficiency trade-offs.

The main suggestions from reviewers concern extending the scope toward robustness under distribution shift, stronger theoretical explanations for low-rank adaptations, clearer benchmarking metadata, standardized evaluation protocols, and broader task coverage beyond image classification. These are largely framed as desirable extensions rather than critical flaws.

In the rebuttal, the authors clearly justify the intended scope of the survey and appropriately position these points as future research directions. They also commit to concrete clarifications regarding benchmarking protocols, efficiency interpretation, and architectural generalization.

Overall, the paper is sound, well-executed, and its limitations are well understood and properly addressed.

Based on the reviews and rebuttal, the recommendation is to accept this paper.

---

### Decision · Program_Chairs · 2026-01-16

Accept (Oral)